# Assessing the Role of Aquaporin 4 in Skeletal Muscle Function

**DOI:** 10.3390/ijms24021489

**Published:** 2023-01-12

**Authors:** Tejal Aslesh, Ammar Al-aghbari, Toshifumi Yokota

**Affiliations:** 1Neuroscience and Mental Health Institute, Faculty of Medicine and Dentistry, University of Alberta, 116 St. and 85 Ave., Edmonton, AB T6G 2E1, Canada; 2Department of Medical Genetics, Faculty of Medicine and Dentistry, University of Alberta, 116 St. and 85 Ave., Edmonton, AB T6G 2E1, Canada; 3The Friends of Garret Cumming Research and Muscular Dystrophy Canada HM Toupin Neurological Science Research Chair, 8812 112 St., Edmonton, AB T6G 2H7, Canada

**Keywords:** aquaporins, AQP4, AQP1, mdx, dystrophin, Duchenne muscular dystrophy (DMD), skeletal muscle, dystrophin-associated protein complex (DAPC), limb-girdle muscular dystrophy (LGMD), dysferlinopathy, sarcoglycanopathy, neuromyelitis optica (NMO), Fukuyama-type congenital muscular dystrophy (FCMD), alpha1-syntrophin

## Abstract

Water transport across the biological membranes is mediated by aquaporins (AQPs). AQP4 and AQP1 are the predominantly expressed AQPs in the skeletal muscle. Since the discovery of AQP4, several studies have highlighted reduced AQP4 levels in Duchenne muscular dystrophy (DMD) patients and mouse models, and other neuromuscular disorders (NMDs) such as sarcoglycanopathies and dysferlinopathies. AQP4 loss is attributed to the destabilizing dystrophin-associated protein complex (DAPC) in DMD leading to compromised water permeability in the skeletal muscle fibers. However, AQP4 knockout (KO) mice appear phenotypically normal. AQP4 ablation does not impair physical activity in mice but limits them from achieving the performance demonstrated by wild-type mice. AQP1 levels were found to be upregulated in DMD models and are thought to compensate for AQP4 loss. Several groups investigated the expression of other AQPs in the skeletal muscle; however, these findings remain controversial. In this review, we summarize the role of AQP4 with respect to skeletal muscle function and findings in NMDs as well as the implications from a clinical perspective

## 1. Introduction

The skeletal muscle is of utmost importance in the osmotic equilibrium of the human body because it holds the largest volumes of both intracellular and interstitial fluids in the body [1]. Under physiological stress during activity, there is a shift in the fluid volumes occurring in and out of the muscle. The force required to facilitate this volume change comes from intracellular osmolyte production associated with mechanical stress or activity [1,2]. Water movement across membranes in mammalian cells occurs by simple diffusion. Unfortunately, this speed is insufficient to promote rapid water fluxes to regulate homeostasis. A family of water channels, including aquaporin-1 (AQP1) and AQP4, plays an important role here.

The passive transport of water, a substance that makes up a substantial percentage of the human body, is undertaken by AQPs down the gradient throughout biological membranes [3]. AQPs are a group of selective integral membrane proteins that facilitate water and non-ionic molecule (i.e., glycerol and urea) transport across the membranes of a diverse group of cells in organisms [4]. Historically, AQP1, found in erythrocytes, was the first to be cloned and purified by Agre and colleagues, and its cDNA was subsequently sequenced [5,6]. This paved the way for later discoveries and a characterization of the water-transport function of AQPs [7,8]. Today, AQPs are functionally categorized into water-only transport channels named orthodox AQPs, and channels that selectively transport water alongside glycerol and urea, also known as aquaglyceroporins. In total, there are 13 known AQPs ranging from AQP0 to AQP12 with AQPs 0, 1, 2, 4, 5, 6, and 8 categorized as orthodox AQPs and AQPs 3, 7, 9, and 10 making up the aquaglyceroporin group [8,9]. AQPs in the membrane form a homotetrameric structure with each monomer containing an independent channel for water transport [3,10]. The membrane-spanning AQPs have an expected weight between 27 to 37 kDa and consist of six transmembrane domains connected via five loops, two of which have half helix components that contain the family’s Asn-Pro-Ala (NPA) motif that lines up the surface of the pore and is involved in hydrogen bond interactions [3].

Given the role of AQPs in regulating water permeability and maintaining homeostasis throughout body tissues and cells, the expression of different AQPs in varying proportions, depending on their localization, serves to help the proper functioning of many tissues [11]. For instance, members of the AQP family, such as AQP1, are expressed in major organs such as the brain to help manage the flow of fluids through processes of secretion and reabsorption, and similarly in kidneys, where AQP1 is involved in water reabsorption necessary for urine concentration [12,13,14,15]. AQPs are also expressed in the eyes, neuromuscular system and have been reported in red blood cells [13]. On the other hand, AQP1 is abundant in the endothelial microvascular system and is expressed in the microvessels of many tissues such as kidneys, lungs, pancreas, salivary glands, and skeletal muscle [16,17,18,19,20,21,22,23]. In the kidney, AQP1 found in the proximal tubules and descending thin limbs of Henle plays a role in the process of water reabsorption and concentration of urine [24,25]. In addition, AQP1 is associated with the formation of cerebrospinal fluid (CSF) [24,26,27]. AQP4 on the other hand is richly expressed in the perivascular astrocytic end-feet projections in the brain and other parts of the central nervous system (CNS) where it is thought to play a crucial role in maintaining the osmotic balance as well as transporting water from blood to the brain [28]. The expression of AQP4 has also been reported in the sarcolemma of skeletal muscles, especially in the fast-twitch fibers [29]. Slow-twitch muscle fibres do not express AQP4 [29].

Despite the growing evidence of AQP1 and AQP4 expression in skeletal muscles, their physiological roles remain elusive. Growing data have also indicated their possible involvement in muscle diseases such as Duchenne muscular dystrophy (DMD). The focus of this review is to present developments in assessing the function of AQP1 and AQP4 in the skeletal muscle, localization, and their role in DMD and neuromuscular disorders.

## 2. Discovery of Orthogonal Array Particles (OAPs)

Freeze-fracture electron microscopy (F-F) identified an assembly of orthogonal arrays (OAs) in the plasma membrane. The F-F technique cleaves the biological membrane into two leaflets known as the protoplasmic (P) face and the extracellular (E) face on which the presence of OAs has been reported in the membranes of a variety of cells including the epithelia of the small intestine, kidneys, skeletal and cardiac muscle cells, satellite cells, etc. [8,30]. The significance of the OA assembly was brought to light by Verkman and colleagues when they discovered its function as a water channel [8,18,31,32,33]. Frigeri et al., 1995, hypothesized the possibility of a connection between OAs and AQP4 channels [8,32]. When Chinese hamster ovary cells were transfected with AQP4 cDNA isolated from a rat, the cells demonstrated the formation of OAs on the P-face which was absent in control cells [33]. This, alongside F-F electron microscopy, provided direct evidence that AQP4 with a unique structure assembles in the OAs [31,34]. Several earlier groups provided evidence supporting the above theory. Verbavatz et al. demonstrated the absence of OAs in an AQP4 knockout murine model, while Rash et al. showed the direct immunogold labeling of AQP4 in the OAs residing in the astrocytes, brain, and spinal cord of a rat [31,35]. In the skeletal muscle, positive anti-AQP4 staining of the OAs in the plasma membrane of rat skeletal myofiber was reported by Shibuya et al. [36].

With respect to the skeletal muscles, the OAs were found to be expressed predominantly in the P face of the plasma membrane, while the E face had pits of OAs [8]. The OA numbers were almost nil at birth, followed by a steady increase until day 27 in normal rat muscle, beyond which the numbers slightly decreased and plateaued [34]. In the murine skeletal muscle, the OA density was highest two months after birth, followed by a gradual decrease [37]. The majority of the human skeletal muscle myofibers demonstrated the presence of OAs on the plasma membrane, while the murine skeletal muscles showed a selective preference for OAs in fast-twitch type 2 myofibers over slow-twitch type 1 myofibers [29,38]. Age-related trends in OA density in humans have not yet been studied.

## 3. Characterizing Aquaporin 4 in the Skeletal Muscle

AQP4 is a mercurial insensitive water channel (MIWC) that was initially cloned from a rat lung and identified as a non-glycosylated protein with six transmembrane domains lacking cysteine at the sites sensitive to mercury [29,39]. AQP4 is most abundantly localized in the neuromuscular system and is expressed in relatively high concentrations in the brain and astrocytes [29,39,40,41].

AQP4 expression was first reported by Frigeri A et al. in skeletal muscle fiber [32]. In skeletal muscles, AQP4 regulates the water flow of myofibers, maintains homeostasis, and is involved in the metabolic and fatigue resistance processes of skeletal muscles [1,29,42,43]. A dramatic difference in the expression levels was observed between the slow and fast-twitch fibers, indicating selective preference of the latter and its subsequent role in exercise and endurance [29]. A study carried out using normal human skeletal muscle biopsies revealed that 64–99% of the muscle fibers were positive for fast myosin heavy chain (MHC) expressed AQP4, while the fibers positive for slow MHC had a variable pattern of AQP4 expression (6–72%). Accounting for the variability, it was interpreted that the correlation between the MHC fast and slow isoforms with AQP4 expression may not always be true [44].

AQP4 exists as two major isoforms differing in length at the N-terminus, shorter AQP4-M23 and longer AQP4-M1, arising from their alternative translation initiation [45]. The ratio of AQP4-M1 and M23 determines the OA number and size. Six other isoforms were also reported in the rat that also arose from alternative splicing [46]. Another AQP4 isoform lacking exon 4 (AQP4 Δ4) was recently identified in humans, arising from the alternative splicing of exon 4. In the majority of the skeletal muscles, the mRNA expression of AQP4 Δ4 correlates inversely with AQP4 protein levels, suggesting a regulatory mechanism through which the cell-surface expression and activity of AQP4 are altered [47]. For example, the EDL muscle, which is a fast-twitch muscle group, had high levels of the protein translated from the full-length AQP4 gene. On the other hand, the flexor digitorum brevis (FDB) muscle, which is an oxidative fast-twitch muscle group, has low AQP4 protein levels due to the higher expression of the AQP4 Δ4 isoform, suggesting its role in the active regulation of AQP4 in the skeletal muscle. The predicted weight of this isoform is 30 kDa, making it indistinguishable from the AQP4-M23 isoform. The translated protein from this isoform localizes intracellularly in the endoplasmic reticulum and not in the plasma membrane. Thus, due to improper trafficking and localization, the protein is rapidly degraded due to its high turnover [47]. The expressions of these isoforms alter depending on the homeostatic requirement of the cells [48].

### Association of AQP4 with the Dystrophin-Associated Glycoprotein Complex (DAPC)

The dystrophin-associated glycoprotein complex (DAPC) has a crucial role in the maturation and maintenance of the skeletal muscle sarcolemma and the neuromuscular junction (NMJ) in vertebrates [49]. The DAPC is made up of dystroglycans and sarcoglycans that span the plasma membrane, out of which α-dystroglycans bind to the extra-cellular matrix (ECM). The cytoplasmic components of the DAPC include dystrobrevin, syntrophin, and dystrophin, binding directly to the cytoskeletal protein actin [49]. AQP4 is predominantly localized to the sarcolemma type II fast-twitch glycolytic fibers and subsequently interacts with the DAPC, mediated by a key player: α-syntrophin (also called α1-syntrophin) [50,51]. The DAPC complex has been illustrated in Figure 1. Due to its close association with α-syntrophin, which in turn facilitates interaction with the DAPC, several groups studied the expression patterns of α-syntrophin and AQP4 in Duchenne muscular dystrophy (DMD), which is caused by mutations in the dystrophin gene.

## 4. Relevance of Aquaporins in Duchenne Muscular Dystrophy (DMD) and Other Neuromuscular Disorders (NMDs)

### 4.1. Duchenne Muscular Dystrophy (DMD)

With a prevalence of 1 in 3500 to 5000 male births every year, Duchenne muscular dystrophy (DMD) is one of the most common inherited neuromuscular disorders affecting the musculoskeletal system [52,53,54]. The causal DMD gene located on the X chromosome is the largest in the human genome, spanning over 79 exons, and is a mutational hotspot for insertions, deletions, mislocations, duplications, point mutations, and complex rearrangements that produce an out-of-frame transcript in DMD patients [53,55,56]. This ultimately leads to the absence of a 427 kDa functional translated protein called dystrophin. The lack of dystrophin results in progressive muscle damage, compromised muscle regeneration, and the replacement of muscle fibers with fibrotic adipose tissue [57,58,59]. As a result, patients are often wheelchair-bound in early adolescence and die prematurely in their twenties due to cardiorespiratory complications [60,61].

The full-length translated dystrophin protein comprises four domains: an actin-binding domain at the N-terminus, a central rod domain made up of 24 spectrin-like repeats (SLRs), a cysteine-rich domain, and a C-terminal domain [53]. Dystrophin protein acts as a stabilizer in the sarcolemma, by bridging the cytoskeletal actin to the extracellular matrix (ECM) with the help of the DAPC complex [62,63]. The absence of dystrophin in DMD destabilizes DAPC moiety, leading to muscle membrane rupture and subsequent necrosis during contractile activity [49,64].

### 4.2. Findings in DMD Models and Patients

Immunofluorescence staining of the skeletal muscle of *mdx* mice, a model for DMD with a nonsense mutation in exon 23, revealed a significant reduction in AQP4 [29,65,66]. Frigeri A et al. hypothesized that this reduction was unrelated to muscle degeneration in DMD. This laid the foundation for several groups to investigate the role of AQP4 in DMD and its involvement in the biochemical alteration of muscle fibers [29].

The *mdx* model is ideal for studying the relevance of AQP4 in DMD pathology. The implication of AQP4 reduction in DMD was studied further using *mdx* mice. Since the CNS is enriched in AQP4, especially in the astrocytic end feet and ependymal cells, it is thought to be a key player in maintaining the osmotic balance of the brain [26,32]. Biochemical analysis of the *mdx* brain showed increased extracellular and decreased intracellular brain volume, but the alterations in AQP4 levels were not reported at that time [67]. Hence, Frigeri et al. investigated AQP4 levels in the *mdx* brain [68]. They reported swollen glial processes, indicative of altered water balance in the mdx brains with a concomitant decrease in AQP4 levels in the dystrophic model. This provides evidence that the role of AQP4 is strictly associated with the reabsorption of water from the extracellular fluid to the blood and CSF [68]. This would possibly result in the slower drainage of water from the brain, resulting in edema and swelling [68]. Densitometric analysis of *mdx* brain homogenates revealed a 30% reduction in AQP4 protein levels in 1-month-old *mdx* mice and 70% in 1-year-old mice compared to age-matched controls [68]. In the tibialis anterior (TA) muscle of 1-month-old *mdx* mice, they observed AQP4 level differences even between the fast-twitch fibers. Type II B fibers were the first to manifest a reduction in AQP4, compared to type II A. In adult *mdx* mice (1-year old), the AQP4 reduction was more general. Interestingly, they observed no differences between *mdx* and control mice at the transcript level in the brain and skeletal muscle, indicating that protein degradation occurs at the posttranslational level [68]. Additionally, they also observed that the reduction in sarcolemmal staining of AQP4 was not accompanied by an increase in the intracellular compartmentalization of the mistargeted protein.

The same group also provided the first evidence of AQP4 reduction in DMD and Becker’s muscular dystrophy (BMD) patients with different mutations in the dystrophin gene [69]. Muscle biopsies from DMD patients aged 6 months-5 years were collected and analyzed for AQP4 expression. AQP4 immunoreactivity was nearly absent in the sarcolemma with no intracellular immunofluorescence. These findings were independent of the age and mutations in the patients [69]. Another interesting aspect of the study was that the reduction in AQP4 levels was not because of a decrease in fast-twitch type IIA muscle fibers, as the patient biopsy revealed several intact fibers [69]. The dystrophic muscles show a predominance of slow-twitch type I fibers that contain fewer OAs. The slow contraction of voluntary muscles in these patients may be related to decreased AQP4 expression [8].

Previously, certain studies suggested the dependence of AQP4 expression on the presence of α-syntrophin [50,70,71]. Upon performing double immunofluorescence on the BMD patient biopsies, Frigeri A. obtained mixed results, arguing that this correlation does not always hold true [69]. However, an α-syntrophin knockout mouse model exhibited an obvious decrease in AQP4 from the sarcolemma, demonstrating the correlation between AQP4 and α-syntrophin expression [50,70,71]. Studies carried out later also revealed that AQP4 reduction is associated with an actual concomitant decrease in α-syntrophin levels in patients [72]. Another group investigated whether the decrease in AQP4 protein levels was a result of reduced mRNA transcripts [73]. Contradicting the mouse model findings, Wakayama et al. observed a marked decrease in AQP4 mRNA expression using quantitative reverse transcriptase PCR (qRT-PCR) in muscle biopsies from DMD patients [73]. Further studies are required to determine the molecular basis of AQP4 reduction in DMD, and the subsequent pathological changes in dystrophic muscle.

### 4.3. Findings in Patients of Other NMDs

In addition to DMD, patients with other NMDs, including Fukuyama-type congenital muscular dystrophy (FCMD), dysferlinopathy, and sarcoglycanopathy, also revealed reduced levels of AQP4, although its roles in pathogenesis and pathophysiology are unclear [74]. FCMD is an early onset childhood progressive muscular dystrophy and is autosomal recessive, caused by mutations in the fukutin gene [75,76]. Although fukutin is ubiquitously expressed, it has stronger expression in the skeletal muscle, heart, pancreas, and brain [75]. It is characterized by joint contracture and mental retardation, cardiomyopathy, and muscle atrophy [76,77,78]. The prevalence rate of FCMD in comparison to DMD is 1:2.1 [76]. The absence of fukutin alters the structure of the basal lamina in the CNS [77,78]. AQP4 expression was found to be significantly reduced in FCMD myofibers. A possible reason for the AQP4 reduction may be attributed to the denervation of the skeletal muscle, which is discussed in the subsequent section.

Studies have also investigated the expression pattern of AQP4 in another group of muscle disorders. Dysferlinopathy includes a spectrum of muscle disorders characterized by two major phenotypes, limb-girdle muscular dystrophy type 2B (LGMDR2) and Miyoshi muscular dystrophy (MMD), characterized by muscle weakness and proximal muscle atrophy [79]. While AQP4 reduction in FCMD correlates with reduced α-syntrophin in the muscle, immunostaining for α1-syntrophin is unchanged in dysferlinopathy patients, suggesting a distinct mechanism involved [80,81].

AQP4 expression was also assessed in patient muscle biopsies affected by sarcoglycanopathies, namely LGMD 2C-F caused by deficiencies of α, β, γ, and δ-sarcoglycan. Sarcoglycanopathies comprise severe forms of autosomal LGMD, with muscle paralysis and wheelchair confinement in early childhood [82]. Knockout mice deficient in sarcoglycans α- γ also showed reduced AQP4 levels [83]. Immunoblot analysis revealed a reduction in overall AQP4 levels, especially the M23 isoform [84]. In conclusion, AQP4 reduction in sarcoglycanopathies is associated with a corresponding decrease in α-syntrophin [51].

## 5. Effects of Exercise on AQP4 Expression in the Skeletal Muscle

AQP4 together with AQP1 has been known to facilitate water exchange between muscle fibers and blood to ensure the volume changes during muscle activity are sustained. This is associated with muscle swelling and intracellular osmolyte production during exercise [85,86,87,88]. To shed light on the relevance of AQP4 in skeletal muscle function, Basco et al. examined the levels of AQP4 in rats when subjected to endurance training [85]. Moreover, they also looked at the contractile properties and endurance performance in mice with ablated AQP4 when compared to wild-type (WT) counterparts. Ten days after endurance activity following treadmill exercise, the levels of AQP4 in the fast-twitch muscles: TA, extensor digitorum longus (EDL), and quadriceps increased significantly compared to day 1 of treadmill activity in rats. However, it was surprising to see the levels not differ significantly on day 30 when compared to day 10 [85]. The increase in AQP4 levels also corroborated the exercise time. For example, the rats that ran on the treadmill for over 30 min exhibited a greater increase in AQP4 compared to the rats that ran for 15 min. This was the first immunoblotting demonstration in which AQP4 accumulation in fast-twitch fibers correlated with the extent of exercise [85]. Thus, the extent of physical endurance determines the increase in AQP4 to keep the water exchange in balance. AQP4 accumulation allows for a rapid change in the fiber volume which is vital to accommodate changes in intracellular concentration and the intracapillary hydrostatic pressure of osmotically active solutes present during prolonged exercise [88].

Basco et al. also investigated the effects of endurance training on the performance in mice deficient in AQP4. When the AQP4 KO mice were subjected to a treadmill-based endurance test, their running capacity was one-third of the WT counterparts on day 1 of the exercise, but they exhibited significant improvement on days 10 and 30, albeit lower than the WT running distance [85]. In a voluntary running wheel exercise, the WT mice still ran a greater distance each day than the KO mice. Nevertheless, the KO mice still showed an improvement in the voluntary running exercise, albeit significantly lower than the WT mice. These experiments demonstrated that ablation of AQP4 impairs muscle strength which ultimately causes the physical performance in daily running to plummet [51]. When they studied the contractile properties of the fast-twitch EDL muscle, they observed no significant difference in the fatigue index, time to peak, or the force generated, indicating that the contraction kinetics of the skeletal muscle was well-preserved [85]. The important takeaway from this study is that AQP4 ablation does not prevent physical activity in mice but limits the KO mice to reach the same performance as their WT counterparts. It also does not hamper the fiber distribution in the muscles. The researchers involved in this study hypothesized that AQP4 ablation did not affect the contractile properties because the duration of the experiment was much shorter, which did not need AQP4-dependent activation, compared to that of the endurance tests where AQP4 accumulation became more evident at day 10 and 30 [85]. 

## 6. AQP1 Acts as a Potential Compensator for AQP4 Loss

AQP1 is the only other aquaporin apart from AQP4 to have confirmed expression in the skeletal muscle. AQP1 expression is enriched in the majority of the tissues, including the blood cells, endothelial, and smooth muscle cells [1,51,89]. In the skeletal muscle, AQP1 is expressed by the endothelial cells in the capillaries arranged between the myofibers, and not on the plasma membrane or the cytoplasm [1]. AQP1 localization in the skeletal muscle fibers was demonstrated by Au et al. and Jimi T et al. [90,91]. Expressions of sarcolemmal AQP4 coupled with vascular AQP1 were thought to facilitate the quick transport of water from the blood into the muscle during intense physical activity [1]. 

Western blot analysis revealed elevated levels of AQP1 in DMD patient biopsies compared to age-matched healthy controls [51]. AQP1 distribution in the sarcolemma was weak in the control muscle cross-sections, but the intensity increased in the DMD biopsies [51]. Unlike AQP4, AQP1 expression was consistent in the sarcolemma of all fiber types. Interestingly, primary cultures of patient-derived human myoblasts showed altered expression of neither AQP4 nor AQP1. Primary cultures are not subjected to mechanical stress or disease modifiers such as increased cytokine expression and are therefore good models to characterize the expression patterns associated with DMD mutations. It is also worthwhile to mention that AQP1 upregulation may not always correlate with AQP4 loss. For example, limb-girdle muscular dystrophy type 2B (LGMDR2) patient biopsies that showed a reduction in AQP4 levels, did not express a concomitant increase in AQP1 [51]. AQP1 KO mice analysis revealed the increased migration of endothelial cells alongside vascular growth, which has implications for increased AQP1 expression in the endothelia of malignant tumors [92,93,94,95]. Since *mdx* mice have also demonstrated enhanced vascular growth, it is possible that upregulated AQP1 expression in DMD endothelia may be attributed to the regenerative capacity of DMD capillaries [90,96]. 

## 7. Effects of Skeletal Muscle Atrophy and Denervation on the Expressions of AQP4 and AQP1

Whenever there is a demand for increased muscle use, AQP4 expression is upregulated to respond to the change in muscle volume and therefore is considered to play a part in the maintenance of muscle hypertrophy [97]. It was recently demonstrated that AQP4 and a non-selective cation channel activated by osmotic stress, called transient receptor potential vanilloid_4_ (TRPV_4_), together regulate cell volume in astrocytes [98,99]. TRPV_4_ is also detected in the skeletal muscle and is therefore thought to also regulate muscle atrophy or hypertrophy along with AQP4 [100,101]. Ishido M. showed that TRPV_4_ accumulation increases in denervation-induced muscle atrophy [102]. Since AQP4 levels are maintained in muscular hypertrophy, it is possible that the balance between AQP4 and muscle volume is impaired in atrophy arising from denervation [97]. Additionally, the same group also provided evidence that AQP4 reduction in the denervated muscle is independent of α1-syntrophin levels. In muscular atrophies, there is a reduction in muscle mass and apparent denervation [103]. Muscle atrophy may be triggered by an interplay of several factors including injury, inflammation, metabolic stress, and glucocorticoid levels [104,105]. This also leads to upregulated levels of Atrogin-1, a marker for denervation [106]. A group reported reduced AQP4 levels in humans and mice with rotator cuff tears (RCT), a form of muscle injury that comprises fatty infiltration and musculoskeletal issues [107]. Muscle atrophy associated with RCT injury in a mouse model also saw a decrease in AQP4 levels. AQP4 loss was attributed to atrogin-1-mediated degradation, mediated by the ubiquitination pathway [108]. It is also worthwhile to mention increased AQP4 levels as a result of atrophy from hindlimb unloading [68,109]. The discrepancy in these findings may be due to the difference in atrophy induction, where the latter technique does not hamper innervation, but the former induces denervation by cutting the sciatic nerve [102]. While neurological disorders such as amyotrophic lateral sclerosis (ALS) show reduced mRNA AQP4 expression, as well as protein levels in the skeletal muscle [110], it would be intriguing to examine these findings in spinal muscular atrophy (SMA), one of the most common genetic neuromuscular disorder affecting infants worldwide [111]. SMA arises from a genetic mutation in the survival of the motor neuron 1 (*SMN1*) gene that causes the loss of motor neurons in the anterior horn of the spinal cord, ultimately leading to the denervation of the skeletal muscle [112,113]. The effects of nerve supply on AQP4 transcription during the early stages of denervation need to be elucidated in the future.

In response to hypertrophy or atrophy, the capillary supply is facilitated or declined [114,115]. AQP1, which is predominantly present in the endothelial cells of the blood capillaries, does not have altered expression post-denervation induced by sciatic nerve freezing [43,116]. Thus, the compensatory nature of AQP1 for AQP4 loss induced by muscle denervation and subsequent hypertrophy may not hold true. The regulation of AQP1 expression occurs independently of the nerve supply to the skeletal muscle [116]

## 8. Expression of Other AQPs in the Skeletal Muscle

To investigate the compensatory roles of AQPs in the skeletal muscle, several groups evaluated the presence of other AQPs. Wang et al. reported the expressions of AQP 3, 4, 5, 7, 8, and 9 in the human masseter muscle, and that of AQP 1, 3, 4, and 10 in the human infrahyoid muscle using RT-PCR [117]. However, the expression of these AQPs was linked to the possible contamination and infiltration by erythrocytes (AQP1), adipocytes (AQP7), and leucocytes (AQP9). To rule out this possibility, Wakayama et al. performed immunocytochemical tests and confirmed the expression of AQP 3, 5, 7, and 9 in the skeletal muscle fibers [4,118,119,120].

AQP3 is a glycerol-transporting integral protein (GLIP) that was cloned from a rat kidney [118,121,122,123]. AQP3 is expressed in the basolateral membrane of collecting duct cells in the kidney, pancreas, small intestine, liver, etc. [124]. Wakayama et al. showed the presence of AQP3 in the plasma membrane of skeletal muscle myofibers in addition to AQP4 using immunoblots and immunostaining, with no apparent preferential expression in the fiber types [118]. The co-expression of AQP3 and AQP4 resorted to the transport of both water and small non-ionic solutes in the skeletal muscle. However, no other studies have demonstrated AQP3 expression in the muscle.

AQP7 and AQP9 are glycerol channels expressed predominantly in the adipocytes and hepatocytes, respectively [4,119]. The weak expression of AQP9 was shown in the myofiber surface membrane [119]. Decreased feeding and increased fasting upregulate glycerol levels that in turn increase AQP7 and AQP9 expression [125]. In the skeletal muscle, the critical role of AQP9 is to facilitate the uptake of glycerol for glucose production during fasting. AQP9 and glycerol kinase mRNA expressions are downregulated by insulin, which is indicative of normal liver processing. However, in endocrine myopathies such as insulin resistance, AQP7 and AQP9 levels are elevated [126,127]. Wakayama et al. demonstrated the presence of AQP7 predominantly in both fast- and slow-twitch myofibers in human skeletal muscle and only in the fast-twitch type II myofibers in murine skeletal muscle [4]. The presence of AQP7 in the skeletal myofiber remains controversial because the immunostaining carried out by another group was negative for AQP7 [128]. Glucose and energy metabolism are essential for energy production in the skeletal muscle, and conditions such as obesity reduce the metabolism, which may induce insulin resistance [129]. Insulin resistance and obesity increase the relative proportion of glycolytic fast-twitch type II fibers but reduce the number of oxidative type I fibers [130,131,132]. Upregulated levels of AQP7 were reported in a mouse model of obesity in the skeletal muscle, indicating its role in insulin resistance [133].

AQP5 was first cloned from salivary glands and was shown to be expressed in the lungs and eyes [134]. Hwang et al. showed the presence of AQP5 in C2C12 myoblasts and that its expression was upregulated upon differentiation into myotubes and also when subjected to hypertonic stress [120]. Western blotting and immunocytochemical staining of C2C12 cells confirmed the presence of AQP5 [120]. The transient upregulation of AQP5 post-differentiation of C2C12 cells may hint at a role in early-stage myogenic differentiation. If AQP5 is a potential compensator for AQP4, there exists a possibility that AQP5 is expressed in the fast-twitch fibers in adult skeletal muscle, and the expression corresponded with the number of fast-twitch fibers [120].

## 9. Concluding Remarks

AQPs facilitate water and non-ionic molecule transport across the membranes of a diverse group of cells in organisms and are therefore essential moieties to be characterized and studied in terms of osmotic regulation and cellular homeostasis. With 13 different AQPs identified so far, there is consensus on the predominance of AQP4 and AQP1 expression in the skeletal muscle. Although there have been a few studies demonstrating the expression of other AQPs such as AQP3, AQP5, AQP7, and AQP9 in the skeletal muscle, these findings are concluded to be controversial. Further evidence may be required to support the results.

AQP4, the most abundantly expressed aquaporin member in the skeletal muscle, is responsible for sarcolemmal permeability, as well as playing a key role in responding to changes in muscle volume during contraction. It has a preferential expression in the sarcolemma of fast-twitch muscle fibers. For the first time, Basco et al. showed a positive correlation between AQP4 levels and endurance activity [85]. An interesting finding put forward by the group was the increase in AQP4 protein levels, and not mRNA, suggesting the role of post-transcriptional regulatory mechanisms in response to cellular stress. On the contrary, Au et al. observed a reduction in AQP4 even at the mRNA level in DMD patients, but not *mdx* mice. Further studies are deemed essential to validate these findings.

When subjected to either forced or voluntary exercise, AQP4 KO mice displayed lesser activity than WT mice. However, there was an improvement in performance after a continued exercise regime. It, therefore, is possible for AQP4 ablation to not prevent activity but limit KO mice from performing similarly to WT mice. This theory was also supported by the contractile studies of the fast-twitch muscle gastrocnemius and EDL muscle, where they observed similar twitch force generation and fatigue resistance in both WT and KO muscles. Additionally, muscle atrophy is not directly associated with AQP4 ablation. Detailed studies are necessary to support this theory.

Sarcolemmal integrity is severely compromised due to dystrophin loss and the subsequent destabilizing of the DAPC complex in DMD patients, coupled with significantly reduced levels of AQP4. Despite being associated with α1-syntrophin, a few studies revealed normal levels of AQP4 in the absence of α1-syntrophin [51]. For example, dysferlinopathy patient biopsies revealed reduced AQP4 levels despite normal α1-syntrophin levels. The water permeability in the sarcolemmal vesicles of *mdx* mice was significantly reduced in one study, with no effect on water permeability in another study. One possible explanation for the permeability remaining unaffected could be the compensatory role of AQP1, which has been demonstrated to be upregulated in *mdx* mice and DMD patients. However, since no increase in AQP1 expression was seen in LGMDR2 patients, this correlation may not always hold true for every muscle pathological disease. Nevertheless, AQP1 probably takes over the role of water transport in the muscle in the absence of AQP4. Future studies can elucidate the role of increased AQP1 in the endothelia of DMD models.

From a clinical perspective, it is important to highlight the pathology presented by patients with neuromyelitis optica (NMO), which is an autoimmune inflammatory disorder characterized by recurring neuritis of the optic nerve [135]. NMO patients produce autoantibodies against AQP4 (AQP4-IgG), leading to reduced AQP4 levels. Suzuki et al. demonstrated elevated levels of CK prior to the onset of optic neuritis (ON) in patients, showing the involvement of skeletal muscle dysfunction in these patients [136]. Physical injury to the skeletal muscle may exacerbate the condition by generating more AQP4-IgG, leading to further loss of AQP4 (). However, minimal changes in muscle pathology were observed in conventional staining. Nevertheless, AQP4 loss and IgG-mediated complement activation on sarcolemma type II myofibers serve as a diagnostic feature in NMO patients [137]. Detailed studies in the future in NMO patients can provide insight into the effect of AQP4 loss on skeletal muscle function.

In conclusion, several studies have highlighted the reduction in AQP4 in muscular dystrophies despite having different causal genetic mutations. The key player responsible for AQP4 reduction is DAPC complex destabilization or loss of α1-syntrophin. Further studies can delve deeper into the relationship of AQP4 loss with the DAPC complex, and validate previous findings. AQP4 and AQP1 are the predominantly expressed AQPs in the skeletal muscle, while the expression of AQPs 3, 5, 7, and 9 remain controversial.

## Figures and Tables

**Figure 1 ijms-24-01489-f001:**
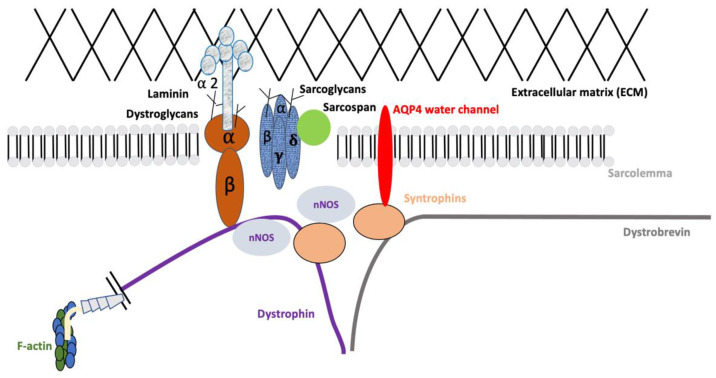
Representation of the DAPC complex in the skeletal muscle. The DAPC complex is a multicomponent system divided into subcomplexes: dystroglycans and sarcoglycans. The cytoplasmic components such as syntrophins and dystrobrevin serve as scaffolds for signalling proteins. AQP4 is localized in the sarcolemma. There are also other signalling molecules (ex. neuronal nitric oxide synthase, nNOS), kinases, ions, etc. Dystrophin, the largest protein in humans, stabilizes the sarcolemma by bridging the cytoskeletal actin to the extracellular matrix (ECM).

## Data Availability

No new data were created or analyzed in this study. Data sharing is not applicable to this article.

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
