# Peer review of "Assessing the Role of Aquaporin 4 in Skeletal Muscle Function"

_ijms, 2023, doi:10.3390/ijms24021489_

Round 1

Reviewer 1 Report

The manuscript is interesting and has merit. I have a few comments.

1.     Much of the text is dedicated to describing AQP function in other tissues than muscle, making the muscle parts which most of the readers will be interested in hard to find. I suggest to make these skeletal muscle less related parts more concise.

2.     Please try to highlight at each subtopic what is still not known about the matter.

3.     Conclusion section is too long and just repeating much of the text from the body of the paper, it should be short, concise and highlighting most important facts and dillemas about AQPs in skeletal muscles.

4.     Most of the references are more than 15 years old. What about the newest literature?   DOI: 10.1038/s41598-020-71167-8 , PMCID: PMC7104575 , DOI: 10.1016/j.jneuroim.2019.577121 , DOI: 10.1080/00207454.2019.1579718

5.     Can AQP7 be also involved in insulin sensitivity of skeletal muscles? ·  DOI: 10.1016/j.redox.2021.102027  , doi: 10.1292/jvms.20-0470 , ·  DOI: 10.1152/ajpendo.00468.2017  ,  DOI: 10.1007/s00018-018-2781-4

Author Response

  1.  Much of the text is dedicated to describing AQP function in other tissues than muscle, making the muscle parts which most of the readers will be interested in hard to find. I suggest making these skeletal muscle less related parts more concise.

We thank the reviewer for their comment and suggestion. Our review predominantly focuses on the skeletal muscle as well as the relevance of AQPs in DMD, a neuromuscular disorder. Nevertheless, we have made the part highlighting aquaporins in tissues other than the skeletal muscle more concise.

  1. Please try to highlight at each subtopic what is still not known about the matter.

We have attempted to highlight the unknown at the end of subtopic.

  1. Conclusion section is too long and just repeating much of the text from the body of the paper, it should be short, concise and highlighting most important facts and dillemas about AQPs in skeletal muscles.

We thank the reviewer for the suggestion. We have updated the same.

  1. Most of the references are more than 15 years old. What about the newest literature? DOI: 10.1038/s41598-020-71167-8 , PMCID: PMC7104575 , DOI: 10.1016/j.jneuroim.2019.577121 , DOI: 10.1080/00207454.2019.1579718

We thank the reviewer for their suggestion. We have added newest literature to the article.

Reviewer 2 Report

The description of the functional role of AQP1 and AQP4 in the skeletal muscle needs to be improved. Currently, the review does well to describe changes with dmd and exercise but it is unclear what the precise role of these proteins are and why their change in expression affects the skeletal muscle with these conditions.

The review should elaborate on other conditions where AQP1 and AQP4 may play a role in skeletal muscle function. 

The scope of the review is very limited in its present form and does not provide a clear and detailed description of the AQP family, especially as it relates specifically to skeletal muscle.

Additional figures need to be added to illustrate the points the authors are making and should be included to help summarize the findings of the studies focused on AQP1/4 expression.

Since this field appears rather limited the concluding remarks should include discussion of the importance of AQPs and where the field needs to go from here. 

Author Response

1. The description of the functional role of AQP1 and AQP4 in the skeletal muscle needs to be improved. Currently, the review does well to describe changes with dmd and exercise but it is unclear what the precise role of these proteins are and why their change in expression affects the skeletal muscle with these conditions.

We thank the reviewer for their suggestion. We have included further evidence and findings that focus on AQP4 and AQP1 expression changes in muscle function. 

2. The review should elaborate on other conditions where AQP1 and AQP4 may play a role in skeletal muscle function. 

We thank the reviewer for their comment. We have included available information on the same. 

3. The scope of the review is very limited in its present form and does not provide a clear and detailed description of the AQP family, especially as it relates specifically to skeletal muscle.

We thank the reviewer for their comment. The main idea of this review was to summarize findings pertaining to the role of AQP4 and 1 in the skeletal muscle, in connection with DMD. There have been only a few studies that have shown reduced AQP4 expression or increased AQP1 in DMD. Nevertheless, we have incorporated additional information pertaining to skeletal muscle denervation and AQP4. We also have included other muscle disorders such as Fukuyama congenital muscular dystrophy and dysferlinopathy, and the findings relevant to AQP4 expression.

Additional figures need to be added to illustrate the points the authors are making and should be included to help summarize the findings of the studies focused on AQP1/4 expression.

The findings that we have included in our review are results from primary articles. Figures could not be reused in the study as we do not have the necessary permissions.  

Since this field appears rather limited the concluding remarks should include discussion of the importance of AQPs and where the field needs to go from here.

We thank the reviewer for this suggestion. We have updated the concluding remarks accordingly.